# A collagen glucosyltransferase drives lung adenocarcinoma progression in mice

Hou-Fu Guo [1], Neus Bota-Rabassedas [1], Masahiko Terajima [2], B. Leticia Rodriguez[1], Don L. Gibbons [1], Yulong Chen[1], Priyam Banerjee[1], Chi-Lin Tsai [3], Xiaochao Tan[1], Xin Liu[1], Jiang Yu[1], Michal Tokmina-Roszyk[4], Roma Stawikowska[4], Gregg B. Fields[4], Mitchell D. Miller [5], Xiaoyan Wang[3], Juhoon Lee[6,7], Kevin N. Dalby[6,7], Chad J. Creighton [8,9], George N. Phillips Jr [5,10], John A. Tainer [3], Mitsuo Yamauchi[2] & Jonathan M. Kurie [1]✉

Cancer cells are a major source of enzymes that modify collagen to create a stiff, fibrotic tumor stroma. High collagen lysyl hydroxylase 2 (LH2) expression promotes metastasis and is correlated with shorter survival in lung adenocarcinoma (LUAD) and other tumor types. LH2 hydroxylates lysine (Lys) residues on fibrillar collagen's amino- and carboxy-terminal telopeptides to create stable collagen cross-links. Here, we show that electrostatic interactions between the LH domain active site and collagen determine the unique telopeptidyl lysyl hydroxylase (tLH) activity of LH2. However, CRISPR/Cas-9-mediated inactivation of tLH activity does not fully recapitulate the inhibitory effect of LH2 knock out on LUAD growth and metastasis in mice, suggesting that LH2 drives LUAD progression, in part, through a tLH-independent mechanism. Protein homology modeling and biochemical studies identify an LH2 isoform (LH2b) that has previously undetected collagen galactosylhydroxylysyl glucosyltransferase (GGT) activity determined by a loop that enhances UDP-glucose-binding in the GLT active site and is encoded by alternatively spliced exon 13 A. CRISPR/Cas-9-mediated deletion of exon 13 A sharply reduces the growth and metastasis of LH2b-expressing LUADs in mice. These findings identify a previously unrecognized collagen GGT activity that drives LUAD progression.

[1] Department of Thoracic/Head and Neck Medical Oncology, The University of Texas MD Anderson Cancer Center, Houston, TX, USA. [2] Division of Oral and Craniofacial Health Sciences, Adams School of Dentistry, University of North Carolina at Chapel Hill, Chapel Hill, NC, USA. [3] Department of Molecular and Cellular Oncology, The University of Texas MD Anderson Cancer Center, Houston, TX, USA. [4] Institute for Human Health & Disease Intervention (I-HEALTH) and Department of Chemistry & Biochemistry, Florida Atlantic University, Jupiter, FL, USA. [5] Department of Biosciences, Rice University, Houston, TX, USA. [6] Division of Medicinal Chemistry, Targeted Therapeutic Drug Discovery and Development Program, College of Pharmacy, The University of Texas at Austin, Austin, TX, USA. [7] Division of Chemical Biology & Medicinal Chemistry, College of Pharmacy, The University of Texas at Austin, Austin, TX, USA. [8] Department of Medicine, Dan L. Duncan Cancer Center, Baylor College of Medicine, Houston, TX, USA. [9] Department of Bioinformatics and Computational Biology, The University of Texas MD Anderson Cancer Center, Houston, TX, USA. [10] Department of Chemistry, Rice University, Houston, TX, USA. ✉email: jkurie@mdanderson.org

Fibrillar collagens play a key role in maintaining tissue integrity and function[1]. Their structural, biochemical, and mechanical properties are regulated, in part, by post-translational modifications of Lys residues[2]. Fibrillar collagens have a central triple-helical structure ("helical domain") and amino- and carboxy-terminal non-helical "telopeptide" domains. Both Helical Lys (hLys) and Telopeptidyl Lys (tLys) residues can be hydroxylated to form helical hydroxylysine (hHyl) and telopeptidyl hydroxylysine (tHyl). tLys and tHyl are oxidized into allysine and hydroxyallysine, respectively, by lysyl oxidases. These aldehydes (allysine and hydroxyallysine) then undergo a series of condensation reactions with allysine, Lys, Hyl, and histidine (His) residues to form collagen cross-links that stabilize collagen fibrils and matrices. Hydroxylation of tLys into tHyl has little impact on cross-link density but, upon oxidation, leads to the formation of a group of structurally distinct stable collagen cross-links called Hyl aldehyde-derived collagen cross-links (HLCCs) that generate tensile strength in load-bearing skeletal tissues like bone and cartilage[2]. Following oxidation and condensation, unhydroxylated tLys forms Lys aldehyde-derived collagen cross-links (LCCs) commonly seen in soft tissues. hHyl residues in collagen can be further modified by a 2-step glycosylation (galactosylation and glucosylation) to regulate how collagens interact with collagen receptors on cells[4–6]. Reduced Lys hydroxylation and/or glyco-sylation underlie inherited connective tissue disorders[7–11], and aberrantly high HLCC production contributes to fibrotic diseases and cancer metastasis[3,12–15].

Key collagen modifications are governed by lysyl hydroxylase family members (LH1-3) that have a conserved dual enzymatic domain architecture but are functionally distinct[15]. While all three LH family members hydroxylate helical Lys (hLys) residues in x-Lys-glycine sequences, LH3 is reportedly unique in its ability to function in tandem with GLT25D1/2 to convert helical Hyl residues into 1,2-glucosylgalactosyl-5-hyl through two consecutive reactions: GLT25D1/2-mediated O-linked conjugation of galactose, and LH3-mediated conjugation of glucose to galactosyl-5-Hyl[16–19]. In addition, LH2 is unique in its ability to hydroxylate tLys residues, leading to the formation of stable HLCCs, as demonstrated by evidence that inactivating mutations of the LH2-encoding gene PLOD2 in Bruck Syndrome are associated with HLCC deficiencies and severe skeletal abnormalities[7,9]. In this study, we sought to elucidate the structural basis for LH2's telopeptidyl LH (tLH) activity and to determine whether tLH activity underlies the pro-metastatic activity of LH2 in LUAD.

## Results

### LH2-dependent LUAD models.
We implemented an immuno-competent LUAD model[20] to identify structural features of LH2 that promote tumor growth and metastasis. Relative to parental K-ras/Tp53-mutant 344SQ LUAD cells, 344SQ cells subjected to CRISPR/Cas-9-mediated Plod2 knockout (KO) generated orthotopic LUADs that were smaller and less metastatic to mediastinal lymph nodes and the contralateral lung (Fig. 1a-c, Supplementary Fig. 1). Compared to parental cells, Plod2 KO 344SQ cells demonstrated no loss of proliferative activity in monolayer culture (Fig. 1d) but generated multicellular aggregates that were less invasive in collagen gels (Fig. 1e, f), which is in line with evidence that LH2 promotes LUAD invasive activity[3].

Enhanced intra-tumoral fibrosis is associated with reduced T cell infiltration, M2 macrophage polarization, and increased recruitment of regulatory T cells and myeloid-derived suppressor cells that inhibit CD8+ T cell immunity[21]. Immune cells express collagen receptors that have immunosuppressive functions[22]. To determine whether LH2 influences intra-tumoral immunity, we quantified immune cell subsets in subcutaneous tumors generated by Plod2 KO or parental 344SQ cells and identified alterations in T cell and myeloid cell subsets that were consistent with an anti-tumor response in Plod2 KO tumors (Fig. 1g, h). Thus, LH2 influences collagen's immunosuppressive functions.

### Electrostatic interactions specify tLH activity.
Given that the LH domain active site is highly conserved across LH family members[23], we reasoned that residues extrinsic to LH2's active site determine tLH activity. To identify the domain in which those residues reside, we performed rescue experiments on LH2-deficient MC3T3-E1 (MC) osteoblasts and found that residues required for HLCC reconstitution reside in LH2's LH domain (Fig. 2a-g, Supplementary Fig. 2). LH2 homology modeling based on a mimiviral tLH domain crystal structure (Fig. 2h) identified 2 basic amino acid residues (R680 and R682) adjacent to the LH2 active site that are dramatically different in charge and hydro-phobicity from the corresponding amino acids in LH1 and LH3 (Fig. 2h, i). Furthermore, highly acidic aspartate and glutamate residues are positioned adjacent to tLys residues on fibrillar collagens (Supplementary Fig. 3). Replacing R680 and R682 on LH2 with the corresponding residues on LH1 (LH2-EP) ablated tLH activity (Fig. 2j), whereas activity on type I collagen, which contains hLys residues, was relatively preserved (Fig. 2k). Conversely, replacing collagen telopeptide's two acidic residues with alanine reduced the activity of wild-type, but not LH1-mimic, LH2 (Fig. 2l, m). These data suggest that LH2's unique tLH activity is determined by electrostatic interactions between LH2 and collagen telopeptides.

### tLH inactivation does not recapitulate the effect of Plod2 KO.
Based on the temporal relationship between stable collagen cross-link accumulations and enhanced tumor cell invasion and metastasis[3,13,24–26], we postulated that tLH activity accelerates LUAD progression. To test this hypothesis, we ablated tLH activity in 344SQ cells by introducing mutations that reduce $Fe^{2+}$-binding in the active site or disrupt LH2 dimer assemblies (Supplementary Fig. 4) without altering LH2 protein levels in 344SQ cells (Supplementary Fig. 5). These mutations reduced orthotopic LUAD metastatic capacity but not size (Fig. 2n, o) and did not fully recapitulate the effect of Plod2 KO (Fig. 1a-c), suggesting that LH2 drives LUAD progression through tLH-dependent and -independent mechanisms.

### LH1 and LH2 have collagen GGT activity.
With respect to potential tLH-independent mechanisms of action, we reasoned that the GLT domain of LH2 may have enzymatic activity that escaped previous detection[27]. By sequence alignment, the LH3 GLT domain has 60 and 57% sequence identity to the corresponding domains of LH1 and LH2, respectively, and the DXXD motif[28] is strictly conserved in LH1 but not LH2; D112 and Y114 in LH3 are replaced with glutamate and phenylalanine, respectively, in LH2 to preserve charge and hydrophobicity (Fig. 3a, Supplementary Fig. 6). To determine whether LH1 and LH2 have galactosylhydroxylysyl glucosyltransferase (GGT) activities, we developed a luciferase assay that detects UDP release following reaction of recombinant LH proteins with UDP-glucose and a synthetic amino acid substrate (galactosyl-Hyl) or deglucosylated type IV collagen. Deglucosylation of type IV collagen was achieved by treatment with a collagen glucosidase, protein-glucosylgalactosylhydroxylysine glucosidase (PGGHG) (Supple-mentary Fig. 7)[29]. Under these conditions, all LH family members had detectable GGT activities that were abolished by mutation of $Mn^{2+}$-binding residues or omission of PGGHG pretreatment (Fig. 3b-h).

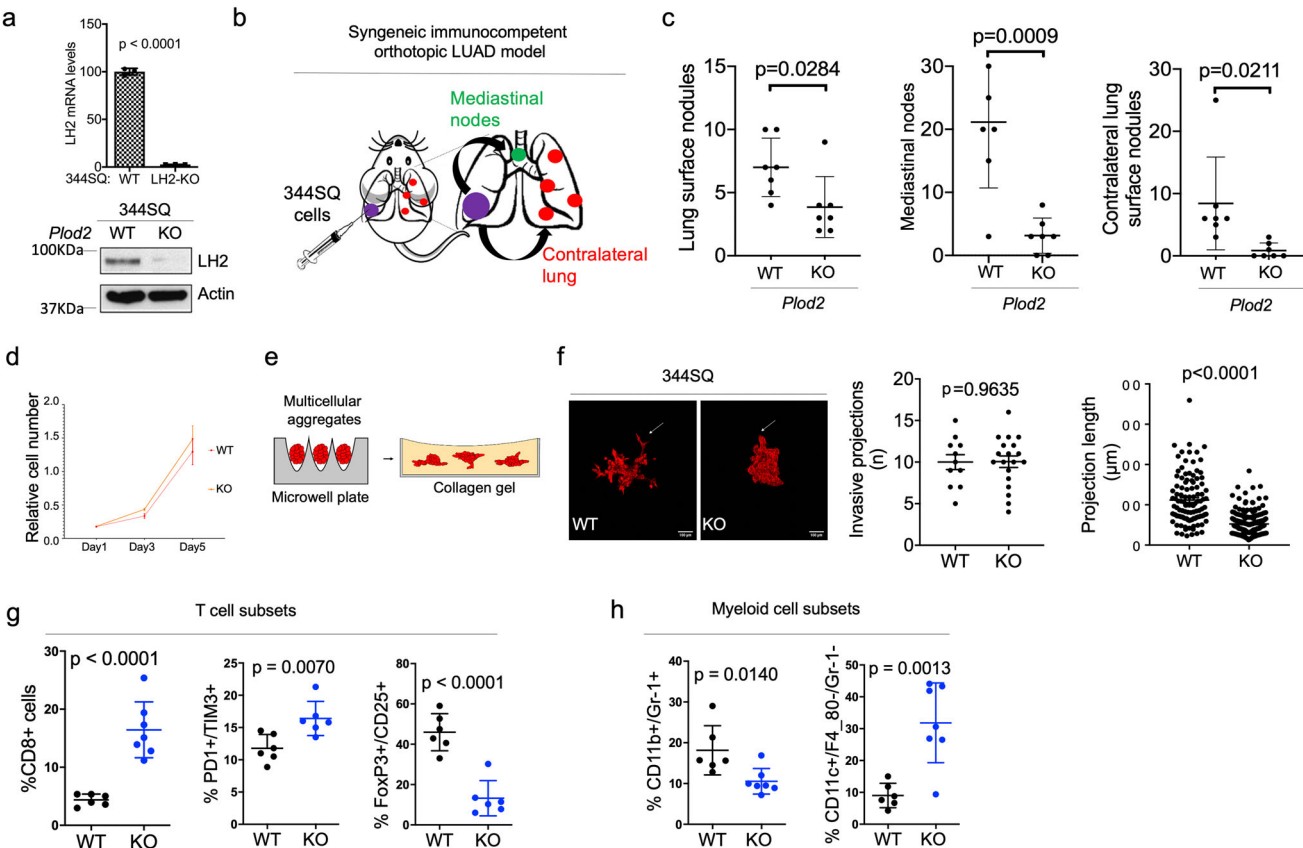

**Fig. 1 LH2 is critical for tumor invasion and metastasis. a** LH2 mRNA levels were measured by quantitative real-time polymerase chain reaction (qPCR) (bar graph, $n = 3$). LH2 protein levels were determined by western blot analysis (gels). Parental (WT) and CRISPR (clustered regularly interspaced short palindromic repeats)/Cas-9 (CRISPR associated protein 9)-mediated *Plod2* knockout (KO) 344SQ cells. Actin used as loading control. **b** Cartoon of metastatic spread in immunocompetent mice bearing 344SQ orthotopic lung tumors. Mice were injected intra-thoracically with parental (WT) or *Plod2* knockout (KO) 344SQ cells to generate a single orthotopic lung tumor. After 2 weeks, mice were necropsied and visible tumors on the pleural surface of the contralateral lung were counted. **c** Numbers of visible primary lung tumors (left dot plot, $n = 7$) and metastases to mediastinal nodes (middle plot, $n = 7$) or contralateral lung (right plot, $n = 7$) per mouse (dot). 344SQ cells described in (**a**). **d** Relative cell densities in monolayer culture determined at each time point by WST-1 assays ($n = 6, 7$ or 8). **e** Cartoon of multicellular aggregates generated in microwell plates, transferred to collagen gels, and examined for formation of invasive projections. **f** Fluorescence micrographs of multicellular aggregates containing RFP-tagged 344SQ cells (images). Invasive projections (arrows). Left plot, number of invasion projections per aggregate (dot, $n = 11$ or 19). Right plot, length of each invasive projection (dot, $n = 110$ or 191). KO aggregates demonstrated reduced invasive projection length. **g** and **h** The percentages of T cells (**g**) and myeloid cells (**h**) in subcutaneous 344SQ tumors were measured by flow cytometric analysis. CD8 + T cells (CD8 +, $n = 6$ or 7), exhausted CD8 + T cells (PD-1 + TIM3 +, $n = 6$), regulatory T cells (FoxP3 + CD25 +, $n = 6$), myeloid-derived suppressor cells (CD11b + Gr-1 +, $n = 6$ or 7), dendritic cells (CD11c + F4_80- Gr-1-, $n = 6$ or 7). Results are mean values (±S.D.) from replicate samples. Error bars indicate ±S.D. $p$ values, 2-tailed Student's $t$ test.

Because key residues in LH3's GLT active site are only partially conserved in LH2, we speculated that LH2's GGT activity has a distinct structural basis. LH2 is alternatively spliced into isoforms that do (LH2b) or do not (LH2a) include exon 13A, which encodes 21 amino acids that are reported to regulate tLH activity[30,31]. However, LH2a and LH2b similarly rescued collagen crosslinking defects in LH2-deficient MC cells (Fig. 3I, j, Supplementary Figs. 8 and 9) and had comparable tLH and hLH activities in enzymatic assays (Fig. 3k, Supplementary Fig. 10). In contrast, GGT activity was sharply higher in LH2b (Fig. 3l, m). To determine how alternative splicing regulates LH2's GGT activity, we modeled LH2b using the recently determined full-length LH3 structure[28] and found that exon 13A adopts a loop conformation that is positioned between α10 and β14 of an accessory domain with a Rossmann fold in close proximity to the GLT active site (Fig. 3n), which led us to speculate that the loop influences substrate binding affinity. By microscale thermophoresis, binding affinity to fluorescein-conjugated UDP-glucose was higher for LH2b than LH2a

(Fig. 3o), and the isoforms demonstrated distinct binding modes (Supplementary Fig. 11a), suggesting that the exon 13A-encoded loop may enhance collagen GGT activity by functioning as a GLT active site cap. Unlabeled UDP-glucose competed with fluorescein-conjugated UDP-glucose for binding to LH2b with an IC$_{50}$ of 30 μM (Supplementary Fig. 11b), suggesting that fluorescein is not involved in binding. Thus, LH2b is a collagen GLT that is regulated by cooperative interactions between tandem Rossmann domains.

**LH2b drives LUAD growth and metastasis.** Glucosylgalactosyl-dihydroxylysinonorleucine (GG-DHLNL) levels were higher in human LUAD than they were in adjacent normal lung tissues (Fig. 4a), warranting studies to determine how collagen glycosylation is regulated in LUAD and its role in LUAD progression. In The Cancer Genome Atlas lung cancer cohorts, which are mostly early-stage tumors, LH2a is the predominant isoform, whereas normal lung tissues have similarly low levels of the 2 isoforms (Fig. 4b-d). In contrast, LH2b levels were equal to or higher than

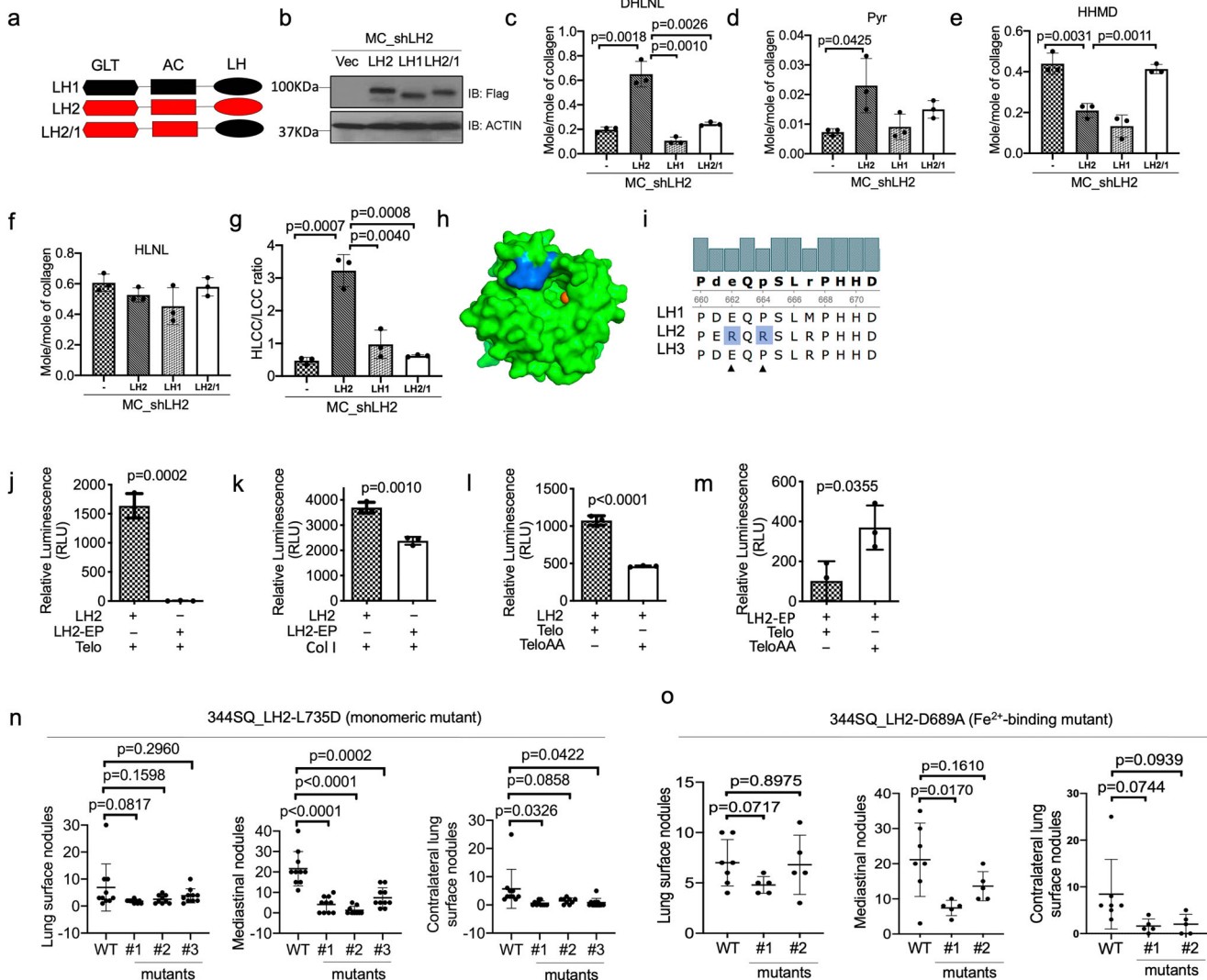

**Fig. 2 LH2 t-LH specificity is determined by electrostatic interactions with collagens. a** Design of LH constructs. Glycosyltransferase (GLT), accessory (AC), and lysyl hydroxylase (LH) domains. LH2's LH domain replaced with that of LH1 (LH2/1). **b** LH2 protein levels were determined by western blot analysis of MC-3T3 (MC) cells that were stably co-transfected with LH2 shRNA (shLH2) and empty vector (-) or vectors expressing LH2, LH1, or LH2/1. Actin used as loading control. **c–g** Collagen cross-link quantification of matrices derived from MC-3T3 (MC) cells described in (**b**). Dihydroxylysinonorleucine (DHLNL, **c**), pyridinoline (Pyr, **d**), histidinohydroxymerodesmosine (HHMD, **e**), hydroxylysinonorleucine (HLNL, **f**), and the HLCC-to-LCC ratio (**g**). The HLCC-to-LCC ratio was calculated as (DHLNL + Pyr)/HHMD. **h** LH2b LH domain structure was modeled using a homology-modelling server SWISS-MODEL. The structure model identifies a cluster of arginine residues (marine) near the LH2 active site. $Fe^{2+}$ molecule (orange ball) in active site. **i** Amino acid sequence alignment of human LHs. R680 and R682 in LH2 (arrow heads) are not in LH1 and LH3. **j–m** LH activity of LH2 and LH2-EP on a trimeric telopeptide (**j, l, m**) or type I collagen (**k**). Acidic residues at i-1 and i-2 positions in telopeptide (Telo) were mutated to alanine (TeloAA). LH activity was measured by detecting succinate production with an adenosine triphosphate (ATP)-based luciferase assay. **n** and **o** Numbers of visible primary lung tumors (left plot) and metastases to mediastinal nodes (middle plot) or contralateral lung (right plot). Mice were injected intra-thoracically with parental (WT) or CRISPR/Cas-9-edited 344SQ cells. *Plod2* mutations ablate LH2's tLH activity owing to loss of dimerization (L735D) (**n**) or $Fe^{2+}$-binding (D689A) (**o**). Results are mean values (±S.D.) from replicate samples. For (**c–g**) and (**j–m**), n = 3. For (**n** and **o**), n = 9 or 10. Error bars indicate ±S.D. *p* values, 2-tailed Student's *t* test.

LH2a levels in human LUAD cell lines and highly, but not poorly, metastatic LUAD cell lines derived from *K-ras/Tp53*-mutant mice (Fig. 4e, Supplementary Fig. 12-14)[20]. Thus, the predominant LH2 isoform switches from LH2a to LH2b during LUAD progression.

Splicing factors that drive exon 13 A inclusion and thereby increase the relative levels of LH2b have been identified[32–34]. One of them, FOX2, is of particular interest because it regulates alternative splicing driven by epithelial-to-mesenchymal transition (EMT)[35,36], which initiates metastasis in *K-ras/Tp53*-mutant LUAD models[20,37–39]. Small interfering RNA-mediated depletion

of FOX2 in 344SQ cells decreased LH2b levels (Fig. 4f, g, Supplementary Fig. 15), indicating that FOX2 promotes exon 13 A inclusion in 344SQ cells.

We subjected 344SQ cells to CRISPR/Cas-9-mediated deletion of exon 13 A to examine the consequences of LH2b loss on LUAD progression. Relative to parental cells, 344SQ_Δexon 13 A cells had no detectable change in total LH2 protein levels (Supplementary Fig. 5) but demonstrated reduced LH2b and increased LH2a mRNA levels (Supplementary Fig. 16). Orthotopic LUADs and flank tumors generated by 344SQ_Δexon 13A cells in syngeneic, immunocompetent mice were smaller and less

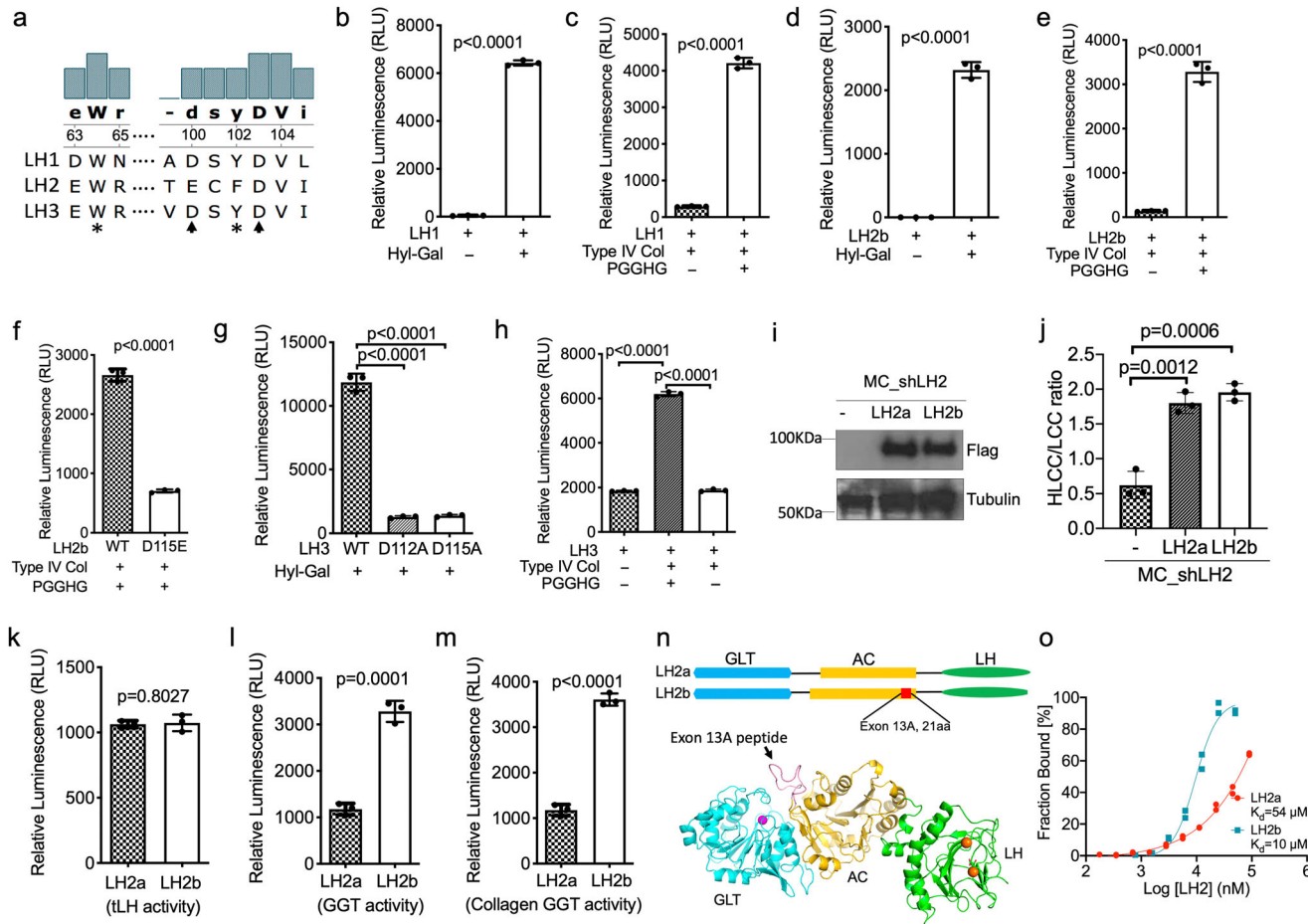

**Fig. 3 LH2b is a collagen GGT. a** Amino acid sequence alignment of human LHs. Residues involved in $Mn^{2+}$- and uridine diphosphate (UDP)-binding are indicated with arrows and asterisks, respectively. **b** and **c** LH1 galactosylhydroxylysyl glucosyltransferase (GGT) activity was assayed. Substrates were hyl-gal (**b**) or type IV collagen (**c**) that had been pre-treated with wild-type (+) or glucosidase-dead mutant (-) protein-glucosylgalactosylhydroxylysine glucosidase (PGGHG). GGT activity was measured by detecting UDP production with an ATP-based luciferase assay. **d–f** Wild type and mutant LH2b GGT activity was assayed using an ATP-based luciferase assay that detects UDP production. Substrates as described in (**b**) and (**c**). GGT activity was abolished by mutation of a $Mn^{2+}$-binding residue (D115E) (**f**). **g** and **h** LH3 GGT assays. Substrates as described in (**b**) and (**c**). Residues involved in $Mn^{2+}$-binding were mutated (D112A, D115A) (**g**). **i** LH2 protein levels were determined by western blot analysis of MC cells stably co-transfected with LH2 shRNA (shLH2) and empty vector (-) or vectors expressing Flag-tagged LH2a or LH2b. Tubulin used as loading control. **j** Quantification of collagen cross-links in matrices produced by MC cells in (**i**). HLCC-to-LCC ratio was calculated as (DHLNL + Pyr)/HHMD. **k** t-LH assay on recombinant LH proteins (LH2a or LH2b) using trimeric collagen telopeptide as substrate. LH activity was measured by detecting succinate production with an ATP-based luciferase assay. **l** and **m** LH2a and LH2b GGT activity was measured by an ATP-based luciferase assay that detects UDP production. Substrates used were hyl-gal (**l**) or PGGHG-treated type IV collagen (**m**). **n** Domain structures of LH2a and LH2b (top). Location of exon 13A-encoded sequences (red bar) in accessory domain (AC). LH2b homology model was generated using a homology-modelling server SWISS-MODEL (bottom). A recently determined LH3 structure was used as a template (PDB ID: 6FXT). Exon 13A-encoded loop (arrow). $Mn^{2+}$ (magenta ball) and $Fe^{2+}$ (orange balls) in GLT and LH domain active sites, respectively. **o** LH2's UDP-glucose-binding affinity was determined by microscale thermophoresis. Fluorescein-conjugated UDP-Glucose (50 nM) was titrated with different concentrations of LH2a and LH2b recombinant proteins to generate the curves. Curves were used to calculate the $K_d$ values for LH2a (red) and LH2b (cyan). Enzymatic activity assay results are mean values (±S.D.) from triplicate samples ($n = 3$) and microscale thermophoresis results are mean values from duplicate samples ($n = 2$). Error bars indicate ±S.D. $p$ values, 2-tailed Student's $t$ test.

metastatic than those generated by parental 344SQ cells (Fig. 4h, i). Multicellular aggregates generated by 344SQ_Δexon 13 A cells were less invasive than aggregates generated by parental or tLH-deficient (L735D) 344SQ cells (Fig. 4j, k). Orthotopic LUADs in nu/nu mice injected with an exon 13A-deficient H358 human LUAD cell line were less metastatic than those generated by parental H358 cells (Fig. 4l), but this difference did not reach statistical significance, potentially owing to low baseline metastatic activity of the orthotopic LUADs or the absence of an intact immune system. The contribution of intratumoral immunity was difficult to ascertain given that the subcutaneous tumors generated by 344SQ_Δexon 13A cells were too small for flow cytometric analysis. Thus, LH2b's GGT activity drives LUAD growth and metastasis.

## Discussion

The presence of a fibrotic tumor stroma is correlated with hypoxia, immunosuppression, treatment resistance, and metastasis[14]. These features result in part from an accumulation of stable collagen cross-links[3,24]. Cancer cells direct the formation of stable collagen cross-links by producing enzymes that hydroxylate (e.g., LH2) or oxidatively deaminate (e.g., lysyl oxidases) Lys residues on collagen[3,24]. Here, we abolished tLH activity by disrupting either $Fe^{2+}$-binding in the active site or

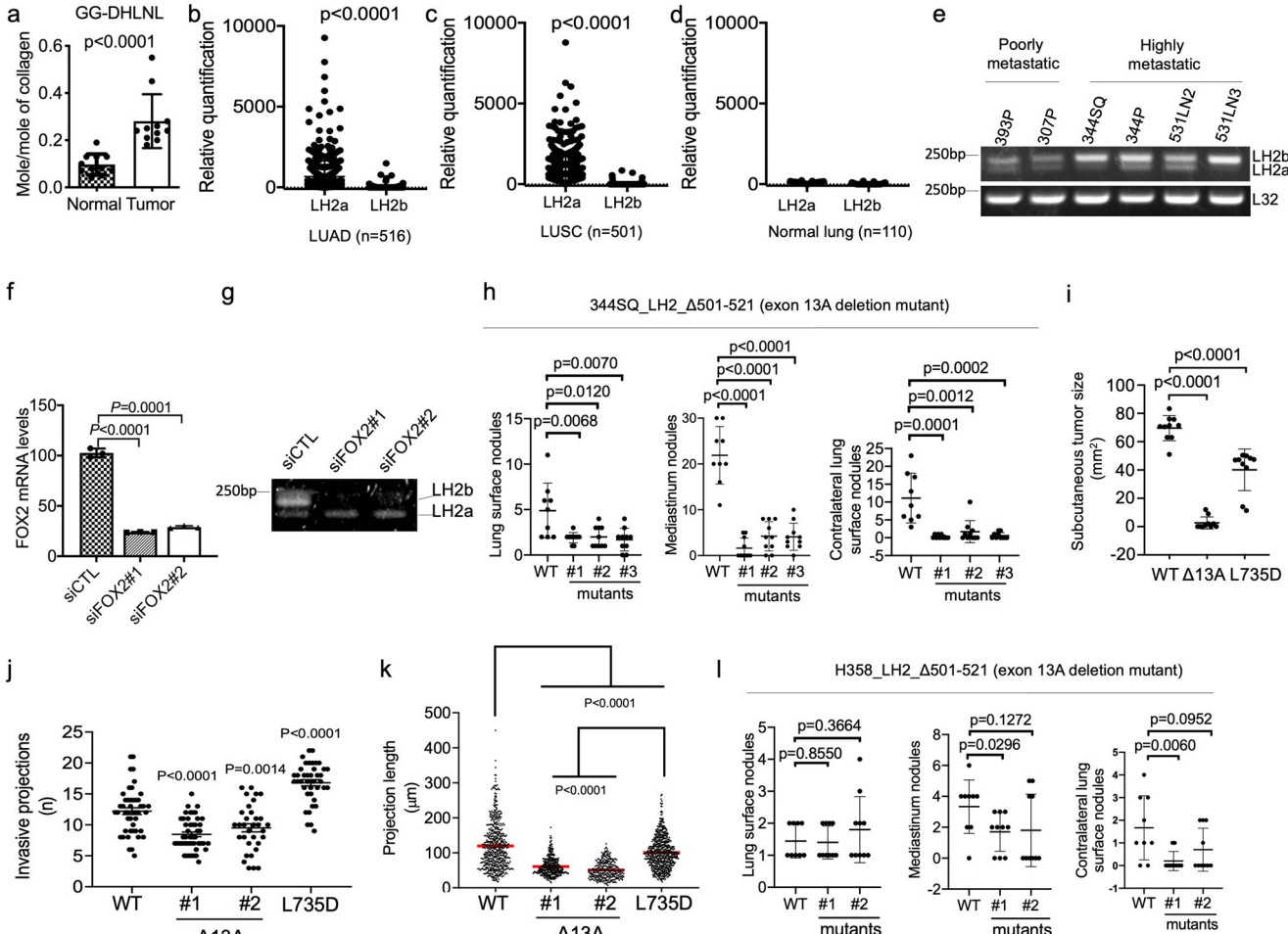

**Fig. 4 LH2b inactivation inhibits orthotopic lung tumor growth and metastasis. a** Quantification of glucosylgalactosyl-DHLNL (GG-DHLNL) in human LUAD and adjacent normal lung samples ($n = 11$ of each). Frozen tissue specimens obtained from Houston Methodist Hospital. **b–d** LH2a and LH2b mRNA levels in tissue specimens (dots) from The Cancer Genome Atlas cohort. LUAD (**b**, $n = 516$), lung squamous cell carcinoma (**c**, $n = 501$), and normal lung (**d**, $n = 110$). **e** Reverse transcriptase PCR (RT-PCR) analysis of LH2a and LH2b mRNA levels in murine K-ras/Tp53-mutant LUAD cell lines. L32 included as loading control. **f** Quantitative RT-PCR analysis of FOX2 mRNA levels in 344SQ cells transfected with control (siCTL) or FOX2 (#1 or #2) siRNAs. Values expressed relative to siCTL-transfected cells ($n = 3$). **g** RT-PCR analysis of LH2a and LH2b mRNA levels in cells described in (**f**). **h** Numbers of visible primary lung tumors (left plot, $n = 9$ or 10) and metastases to mediastinal nodes (middle plot, $n = 9$ or 10) or contralateral lung (right plot, $n = 9$ or 10) per mouse (dots). Mice were injected intra-thoracically with parental (WT) or CRISPR/Cas-9-edited 344SQ cells with LH2 exon 13 A deletion. **i** Subcutaneous tumor size in mice (dots, $n = 10$). Tumor area (length times width) determined at time of sacrifice, which was 3 weeks, 6 weeks and 4 weeks after injection of parental, Δexon 13 A, and L735D 344SQ cells, respectively. **j** Number of invasion projections per multicellular aggregate (dot) ($n = 33$ or 40 or 46 or 47). Multicellular aggregates were created in laser-ablated microwells, embedded in collagen, cultured for 3 days, stained, and imaged. Length of each invasive projection (dot) (**k**, $n = 314$ or 398 or 562 or 672). Mutants include exon 13 A deletion (Δ13 A) or loss of dimerization (L735D). Projection length in Δexon 13 A aggregates was less than that of parental cells and L735D mutants. **l** Numbers of visible primary lung tumors (left plot, $n = 9$ or 10) and metastases to mediastinal nodes (middle plot, $n = 9$ or 10) or contralateral lung (right plot, $n = 9$ or 10) per mouse (dots). Parental (WT) or CRISPR/Cas-9-edited H358 cells with LH2 exon 13A deletion. Results are mean values (±S.D.) from replicate biological samples. Error bars indicate ±S.D. p values, 2-tailed Student's t test.

LH2 dimer assembly formation and found that LUAD progression was minimally impaired. Homology modeling and biochemical studies identified collagen GGT activity that results from an alternatively spliced exon in LH2. These findings identify a previously unrecognized collagen GGT activity that may enable cancer growth and metastasis.

Collagen glycosylation has a profound impact on collagen fibrillogenesis, cross-link maturation, collagen stability, matrix mineralization, axonal guidance, and platelet activation[40–44]. Glycosylation influences the ability of collagens to function as ligands for receptors (e.g., DDRs, integrins, CD44) that direct cellular functions during embryonic development and tissue repair and are aberrantly expressed in numerous pathologic

conditions, including cancer[4–6]. Based on our finding that LH1 and LH2 have collagen GGT activity, we conclude that all LH family members are bifunctional enzymes that function within an integrated collagen regulatory network, and future studies should readdress polymorphisms and inherited mutations in GLT domains of the PLOD gene family that were previously thought to be non-contributory in individuals with connective tissue disorders[45,46]. Moreover, our finding that Hyl glucosylation is a pro-tumorigenic collagen modification opens new avenues of research into how collagen glucosylation influences pro-metastatic processes in the tumor microenvironment.

It remains unclear whether LH2- and LH3-catalyzed GGT activities play distinct or overlapping biochemical and biological

roles[18]. LH2 and LH3 may glucosylate distinct G-Hyl residues on collagen and/or modify different collagen types. The finding that LH2's GGT activity is determined by an alternatively spliced exon raises the possibility that its dual enzymatic functions are coordinately regulated. Such coordinated activities could profoundly influence angiogenesis and other LH family-dependent processes in the tumor microenvironment as recently reported[47]. Although further studies are necessary to dissect the molecular events that link collagen GGT activity to cancer progression, our data raise the tantalizing possibility that drugs that target LH2's GLT active site could be useful for the treatment of LUAD and other LH2-dependent cancer types.

Finally, by performing domain swapping and rescue experiments on LH2-deficient osteoblasts, we showed that LH2's LH domain determines its tLH activity. Site-directed mutagenesis and enzymatic activity assays identified residues that are critical for LH2's tLH activity. These results indirectly suggest that a unique arginine cluster near LH2 active site may directly engage acidic residues in collagen telopeptides to facilitate substrate recognition. However, protein crystallographic studies will be required to directly substantiate this possibility.

## Methods

**Plasmids.** Full-length murine PGGHG was cloned into a modified pET-28b (Novagen) vector using NheI and NotI cloning sites with standard PCR-based methods. This modified pET-28b vector has NheI inserted in the linker between His$_6$ and Thrombin recognition site, which changes the amino acid linker from GS to AS. For recombinant protein production in Chinese hamster ovary (CHO) cells, truncated human LH1 (aa22-727) and LH3 (aa32-738) were cloned into BamH1 and Not1 sites of pSGHP1, which was a generous gift from Dr. Craig W. Vander Kooi (University of Kentucky). Truncated human LH2 (residues 33-737 for LH2a and residues 33–758 for LH2b) were cloned into XbaI and Not1 sites of pSGHP1. Point mutant constructs were generated using QuickChange Lightning Site-Directed Mutagenesis Kit (Agilent). For ectopic expression in MC cells, murine LH1, LH2a, LH2b and mutant were cloned into XbaI and NotI cloning sites of pEF-bsr with standard PCR-based methods[48]. The identities of all constructs used in this study were confirmed by sequencing. Primers used for cloning and mutagenesis are listed (Supplementary Table 1).

**CRISPR/Cas-9 Plod2 editing.** 344SQ and H358 cells were cultured in Roswell Park Memorial Institute 1640 supplemented with 10% FBS (complete media) in a humidified atmosphere with 5% CO$_2$ at 37 °C. To generate Plod2 KO 344SQ cells, we re-constructed the Cas9-2A-GFP vector (Sigma) to express guide RNAs that flank exon1 of Plod2 under the U6 promoter (Supplementary Table 2). 344SQ cells were transiently transfected with vector using lipofectamine 2000 transfection reagent (Thermo Fisher). Three days later, the top 5% of GFP$^+$ cells were isolated by flow sorting and plated as single cells by limiting dilution method. To screen the clones, genomic DNA was extracted from the cells using QuickExtract™ DNA Extraction Solution (Epibio, Inc) and amplified by PCR with PCR primers flanking the deleted region. The deletion was confirmed by quantitative RT-PCR analysis of RNA and Western blot analysis of cell lysates.

For D689A and L735D knock-ins, cells were electroporated with 7 µg all-in-one Cas9/gRNA vector and 0.3 nmol ssODN donor template (Supplementary Table 2). For exon 13 A deletion, cells were electroporated with 10 µg all-in-one Cas9/gRNA vectors. Electroporated cells were allowed to recover for 48 h before being sorted for GFP$^+$ cells and were then plated by limiting dilution at <1 cell per well into 96-well plates in complete medium. Once clones grew to acceptable sizes, they were expanded into 24-well plates and then processed to extract mRNA and genomic DNA for PCR amplification. PCR products were then digested using the SalI enzyme to identify D689A Knock-in clones or digested using the SmaI enzyme to identify L735D Knock-in clones or subjected to gel electrophoresis to identify exon 13A-deleted clones. D689A and L735D knock-in clones were further confirmed by Sanger sequencing.

**Tumor models.** Immunocompetent 129/Sv mice syngeneic to 344SQ cells were bred in-house and randomized to balance the cohorts based on age. Nude mice were purchased from The University of Texas MD Anderson Cancer Center ERO department. Mice were placed under general anesthesia (ketamine/xylazine 50 mg kg$^{-1}$ and 5 mg kg$^{-1}$, respectively, delivered by intraperitoneal injection) and an incision was made on the thorax under sterile conditions to expose the left lung. Tumor cells (10$^6$) were injected directly into the left lung in 50 µl sterile PBS. The incision was closed using staples. The mice were humanely killed after 8 days (344SQ exon 13A-deleted mutants) or 7 days (344SQ LH-inactive mutants) or 7 weeks (H358 exon 13A-deleted mutants) post injection. Metastatic tumors visible on the surface of mediastinal lymph nodes or the right lung were manually

counted. Investigators were blinded to the cohorts at the time of assessment of metastatic tumor numbers. Mice were excluded from the analysis if they died at the time of tumor cell injection due to hemorrhage or pneumothorax. For flow cytometry analysis, syngeneic 129/Sv mice received subcutaneous injections of parental or CRISPR/Cas-9-edited 344SQ cells (1 × 10$^6$ per mouse) in the right flank. The mice were monitored for tumor growth and euthanized 3 weeks after the injection or at the first sign of morbidity. They were necropsied to isolate the primary tumors for flow cytometric analysis. Measurements were taken from a single tumor generated in each mouse (n = 8–10 mice per cohort).

**Flow cytometry.** Tumors were dissociated using the MACS (Miltenyi) mouse tumor dissociation kit. Mechanical dissociation using a gentleMACS Octo dissociator (Miltenyi) was performed followed with enzymatic digestion with collagenase I (0.05% w/v, Sigma), DNase type IV (30 U/ml, Sigma), and hyaluronidase type V (0.01% w/v, Sigma) for 40 min with rocking at 37 °C. Tumor samples were mechanically dissociated again and passed through a 70 µm filter before being stained with fluorochrome-conjugated antibodies in FACS buffer. RBC lysis (Biolegend) was performed on both single cell tumor and splenocytes samples following manufacturer recommendation. Cells were stained for surface markers using fluorochrome-conjugated anti-mouse antibodies (CD45, CD3, CD8, CD4) for 1 h at room temperature. Ghost aqua BV510 (Tonobo) was used to stain dead cells. Cells were fixed using 1% PFA at room temperature for 15 min, then washed twice with perm/wash buffer (Biolegend). Cells were stained with primary antibodies at room temperature for 1 h. Cells were filtered and analyzed on BD LSR Fortessa (BD Biosciences) and analyzed using FlowJo software (v.10.5.3; Tree Star). For single color compensation, ultracomp eBeads compensation beads (Thermo Fisher) were used and stained with a single fluorescent-conjugated antibody according to manufacturer's instructions. Compensation was calculated automatically using BD FACSDiva 8.0.1. Gating schemes utilized for flow cytometry analysis of T cell subsets and myeloid cells are shown in Supplementary Figs. 17 and 18. The antibodies used (clone, dilution, company, catalogue #) are as follows: CD8 PE-Cy7 (53-6.7) 1/800FCBioLegend/100721, CD3 PE-594 (17A2) 1/100 FCBioLegend/100246, CD4 APC-Cy7 (RM4-5) 1/100 FCBioLegend/100526, FoxP3 PerCp-Cy5.5 (FJK-16s) 1/100 FCeBioscience/45-5773-82, CD45 Pacific Blue (30-F11) 1/100 FCBioLegend/103126, CD25 BUV395 (PC61) 1/100 FCBD Biosciences/564022, CD11c BV786 (N418) 1/100 FCBioLegend/117335, GR1 BV711 (RB6-8C5) 1/100 FCBioLegend/108443, TIM3 APC (B8.2c12) 1/100 FCBioLegend/134007, PD1 BV605 (29 F.1A12) 1/100 FCBioLegend/135220, F4/80 APC (BM8.1) 1/100 FCTonbo/20-4801-U100, CD11b BV650 (M1170) 1/100 FCBioLegend/101239, Ghost aqua BV510 1/50 FCTonbo.

**Cell proliferation.** Viable cell densities were quantified in subconfluent culture conditions using water-soluble tetrazolium salt 1 (WST1) reagent as suggested by manufacturer's instructions (Takara). Measurements were taken from distinct samples. Mean values determined from replicate (n ≥ 3) biological samples.

**siRNA knockdown.** FOX2 siRNAs were purchased from Sigma (SASI_Mm02_00305828 and SASI_Mm02_00305829). 344SQ cells were transfected with 100 nM control siRNAs or siRNAs against FOX2. Total RNAs were extracted from the transfected cells 48 h later and cDNA were generated from the RNA samples using qScript cDNA SuperMix (QuantaBio). The expression levels of PLOD2 isoforms were determined using Real-Time PCR.

**Multicellular aggregates.** Multicellular aggregates were created in a 24-well plate containing 1700 laser-ablated microwells per well as described[49]. The microwells were passivated with 0.05% pluronic acid for 1 h prior to seeding the cells. 85,000 LUAD cells were seeded per well and cultured for 48 h to generate aggregates, each containing 50 LUAD cells.

To examine invasive projection formation on multicellular aggregates in collagen gels, multicellular aggregates were mixed with rat tail-derived type I collagen to generate collagen gels with defined concentrations (2 mg ml$^{-1}$), volumes (200 µl per gel), and aggregate densities (35 aggregates per gel). The gels were allowed to polymerize upside-down on a glass-bottom 35 mm dish at 37 °C for 30 min. The aggregates were cultured for up to 3 days, fixed and stained with phalloidin for visualization, and imaged with a NikonA1 confocal microscope, 10× objective. Invasive projections were defined as at least one visible LUAD cell protruding out of the aggregate. The length of the projections was manually quantified using the hand free tool to follow the projection shape (ImageJ). Measurements were taken from distinct samples. Mean values determined from replicate (n ≥ 3) biological samples.

**Protein expression and purification.** PGGHG was expressed in E. coli strain Rosetta (DE3). Cells expressing PGGHG were induced with 1 mM isopropyl β-D-1-thiogalactopyranoside (IPTG) for 16 h at 16 °C. Cells were collected, pelleted and then resuspended in binding buffer (20 mM Tris, pH 8.0, 200 mM NaCl and 15 mM imidazole). The cells were lysed by sonication and then centrifuged at 23,000 g for 15 min. The recombinant PGGHG proteins (wild type or D300E inactive mutant) were purified with immobilized metal affinity chromatography.

Human LH1-3 recombinant proteins were purified from CHO cell–derived conditioned medium samples as described previously[50]. In brief, LH1-3 recombinant proteins were transiently transfected in new Gibco™ ExpiCHO™ cells (Thermo Fisher Scientific, Waltham, MA) with polyethylenimine and expressed as a secreted protein with N-terminal His$_8$ and human growth hormone (hGH) tags via large-scale suspension culture. The LH1-3–containing conditioned medium samples were harvested by centrifugation at 7000 *rpm* for 10 min, filtered through 0.22 μm EMD Millipore Stericup™ Sterile Vacuum Filter Units (EMD Millipore, Billerica, MA), concentrated to 100 mL, and buffer-exchanged into Nickel-binding buffer (20 mM Tris, 200 mM NaCl, 15 mM imidazole, pH 8.0) using the Centramate™ & Centramate PE Lab Tangential Flow System (Pall Life Sciences, Ann Arbor, MI). The recombinant LH proteins were then purified with tandem immobilized metal affinity chromatography and anion exchange chromatography.

**Structure modelling**. Human LH2 LH domain and LH2b full length structure homologies were generated by the SWISS-MODEL (Swiss Institute of Bioinformatics, Biozentrum, University of Basel, Switzerland) homology server[51]. PDB entry 6AX7 and 6FXT were utilized as templates to model LH2 LH domain and LH2b full length, respectively.

**LH enzymatic activity assay**. LH enzymatic activity was measured using a luciferase-based assay as described[50]. In brief, the assay was performed in LH reaction buffer (50 mM HEPES buffer pH 7.4, 150 mM NaCl) at 37 °C for 1 h with 1 μM LH enzymes, 10 μM FeSO4, 100 μM 2-OG, 500 μM ascorbate, 1 mM dithiothreitol, 0.01% triton x-100, and 1 mM wild type (LSYGYDEKSTGGISVP (GPO)$_8$) or mutant (LSYGYAAKSTGGISVP(GPO)$_8$) collagen telopeptide mimics or 4 μM bovine skin collagen substrate containing no telopeptides (Bovine Pure-Col®, Advanced BioMatrix). Collagen telopeptide mimics were dissolved in reaction buffer and incubated overnight at 4 °C to facilitate the formation of trimers, which was confirmed by circular dichroism spectroscopy (Supplementary Fig. 19). Except for LH recombinant protein and bovine skin collagen, all reagents were prepared immediately before use. All these reagents were dissolved in reaction buffer except for FeSO$_4$ and collagen, which was prepared in 10 mM HCl, and the pH of the reaction mixture was checked with pH papers to ensure that HCl did not change the overall sample pH. Bovine skin collagen was denatured by heating at 95 °C for 5 min and then chilled immediately on ice before use. LH activity was measured by detecting succinate production with an adenosine triphosphate–based luciferase assay (Succinate-Glo™ JmjC Demethylase/Hydroxylase Assay, Promega, Madison, WI) according to manufacturers' instructions. Experiments were performed in triplicate from distinct samples, and an unpaired *t*-test was used to compare the enzymatic activity of different samples.

**Circular dichroism spectroscopy**. LH2 recombinant proteins or synthetic telopeptide mimics were analyzed in 0.01 M sodium phosphate and 150 mM NaCl (pH 7.4) at a concentration of 0.5 mg ml$^{-1}$. Circular dichroism spectra were measured using a J-810 spectropolarimeter (Jasco, Easton, MD) with a 2 mm path length quartz cuvette. All measurements were performed at 20 °C and three scans averaged for each spectrum. A blank spectrum of phosphate-buffered saline was collected in the same manner and used for background subtraction. Results represent the mean values from triplicate technical repeats in a single experiment. Each protein was analyzed twice.

**Type IV collagen deglucosylation**. Human type IV collagen (MilliporeSigma, St. Louis, MO) was deglucosylated in deglucosylation reaction buffer (50 mM acetate buffer pH5.3, 150 mM NaCl) at 37 °C for 4 h with 100 ug of PGGHG enzymes and 2500 μg type IV collagen. Type IV collagen treated with inactive PGGHG D300E mutant served as a negative control. After incubation, the reaction was stopped by incubating at 98 °C for 3 min. The deglucosylated type IV collagen (dgCol4) production was indirectly detected by measuring glucose release using Glucose Colorimetric/Fluorometric Assay Kit (MilliporeSigma, St. Louis, MO) according to manufacturers' instructions. Experiments were performed in duplicate from distinct samples, and an unpaired *t*-test was used to compare the enzymatic activity of different samples.

**GGT enzymatic activity assay**. GGT activity was measured in reaction buffer (100 mM HEPES buffer pH 8.0, 150 mM NaCl) at 37 °C for 1 h with 1 μM LH enzymes, 20 μM MnCl$_2$, 100 μM UDP-glucose (MilliporeSigma, St. Louis, MO), 1 mM dithiothreitol, 0.02% bovine serum albumin, and 1 mM galactosyl hydroxylysine (Cayman Chemical, Ann Arbor, MI) or 2 μM dgCol4. GGT activity was measured by detecting UDP production with an ATP-based luciferase assay (UDP-Glo™ Glycosyltransferase Assay, Promega, Madison, WI) according to manufacturers' instructions. Experiments were performed in triplicate from distinct samples, and an unpaired *t*-test was used to compare the enzymatic activity of different samples.

**Western blot**. Cells were washed with PBS and lysed with cell lysis buffer (Cell Signaling Technology, Danvers, MA) to extract total proteins. Cell lysates were separated by SDS-PAGE, transferred onto nitrocellulose transfer membrane using Trans-Blot Turbo Transfer System (Bio-Rad), and then incubated with primary

antibodies and horseradish peroxidase-conjugated secondary antibodies (Proteintech, Sigma and GE Healthcare). Protein bands were visualized using Pierce ECL Western blotting substrate (Thermo Fisher Scientific). Experiments were performed in triplicate from distinct samples. Results are representative of replicate experiments.

**Microscale thermophoresis**. Glucose-UDP-(PEG)$_6$-Fluorescein Conjugate (10 μl at 50 nM, Sigma) was mixed with equal volume of serially diluted unlabeled LH2 proteins in 20 mM HEPES, pH 7.4, 150 mM NaCl, 5 mM Mn$^{2+}$, 0.05% Tween-20. After incubation at 25 °C for 15 min, the samples were loaded into silica capillaries (Nanotemper Technologies). For the competition assay, fixed concentrations of fluorescein conjugated UDP-Glucose (50 nM) and LH2b (20 μM) were titrated with different concentrations of unlabeled UDP-Glucose. Measurements were performed at 20 °C using Monolith NT.115 (Nanotemper Technologies). Data were analyzed (Nanotemper Analysis software. v.1.2.101) to fit K$_d$ according to the law of mass action and to determine IC$_{50}$. The experiment was repeated once. Results represent the mean values from duplicate biological samples.

**Collagen cross-link analyses**. For collagen cross-link analysis, MC cells were cultured for 2 weeks as described. The cell/matrix layer was washed with cold PBS, scraped, collected and pelleted by centrifugation at 10,000 *rpm* for 30 min. The pellets were washed with cold PBS and distilled water, lyophilized, weighed and aliquoted. Aliquots were reduced with standardized NaB$^3$H$_4$, acid hydrolyzed and subjected to amino acid and cross-link analyses as reported[52]. The reducible cross-links, dehydro (deH)-dihydroxylysinonorleucine/its ketoamine, deH-hydroxylysinonorleucine/its ketoamine and deH-histidinohydroxymerodesmosine (for cross-link chemistry, see[2]) were analyzed as their reduced forms, i.e., DHLNL, HLNL and HHMD, respectively, and the mature trivalent cross-links pyridinoline and deoxypyridinoline were simultaneously quantified by their fluorescence. All cross-links were quantified as mol/mol of collagen based on the value of 300 residues of hydroxyproline per collagen molecule. The Hyl content in collagen was calculated as Hyl/Hyp X 300. Experiments were performed in triplicate from distinct samples. Mean values determined from replicate (*n* = 3) biological samples. Because the O-glycosidic linkage of the carbohydrate remains intact in base hydrolysis, the glycosylated immature bifunctional cross-link (GG-DHLNL) was analyzed by subjecting human LUAD to base hydrolysis with 2 N NaOH and analyzed as described previously[18].

**Statistics and reproducibility**. Measurements were taken from replicate biological samples. Results are representative of replicate experiments. Mean values were determined from replicate (*n* ≥ 3) biological samples. Statistical significance was determined using 2-tailed Student's *t* test. Whenever possible, investigators were blinded to the treatment groups at the time of assessment.

**Study approval**. All mouse studies were approved by the Institutional Animal Care and Use Committee at The University of Texas MD Anderson Cancer Center. The use of lung tissues quantification of collagen cross-links in this study was performed under Institutional Review Board–approved protocol IRB(2)0910-01565x at Houston Methodist Research Institute, and written informed consent was obtained from participants or their guardians.

**Reporting summary**. Further information on research design is available in the Nature Research Reporting Summary linked to this article.

## Data availability
Source data underlying plots shown in figures are provided in Supplementary Data 1. All other data, if any, will be available upon reasonable request.

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

## Acknowledgements

This work was supported in part by National Institutes of Health grants R01CA105155 (J.M.K. and M.Y.) and K99CA225633 (H-F.G.); Cancer Prevention and Research Institute of Texas (CPRIT) grant RP160652 (J.M.K); and the Gloria Lupton Tennison Distinguished Professorship in Lung Cancer (J.M.K.).

## Author contributions

J.M.K. and H-F.G. conceived the study. J.M.K. oversaw the work of H-F.G., N.B-R., Y.C., X.L., J.Y., and X.T. H-F.G. designed, performed, and analyzed amino acid sequence alignment and LH and GLT enzymatic activity assays. M.T. and M.Y. designed, performed, and analyzed the collagen cross-linking assays. N.B-R and P.B. designed, performed, and analyzed the studies on 344SQ multicellular aggregates co-cultured on MC cells. B.L.R. and D.L.G performed and analyzed the flow cytometry of immune cell populations. Y.C. and X.T. created expression vectors and performed transfections on MC cells. C-L.T., J.A.T, M. D.M., and G.N.P. assisted with interpretation of protein crystallographic analyses that were removed during manuscript revision. M.T-R., R.S., H-F.G., and G.B.F. prepared substrates utilized in enzymatic assays. X.L. and J.Y. performed mouse breeding, tumor cell injections, and necropsies of orthotopic tumor-bearing mice. K.N.D and J.L contributed to the homology model of LH2 and telopeptide that was not included in the paper.

## Competing interests

The authors declare no competing interests.
