## [Peer Review File · Communications Biology]

Reviewers' comments:

Reviewer #1 (Remarks to the Author):

This manuscript by Guo et al. investigates non-canonical roles of the collagen lysyl hydroxylating enzyme LH2. The authors identify a novel Glucosyltransferase activity of LH2 and show that this function is critical for LH2/PLOD2 mediated invasion and metastasis. The paper is generally well done and shows an important new role for a relatively understudied enzyme, describing its specificity and unique functionality among the other LH isoforms. There are a couple of issues in my opinion which should be resolved prior to publication.

1. All cell line and in vivo work employ a genetically engineered model of lung cancer in the mouse.

It would be helpful to see validation of the physiological relevance of LH2 glucosyltransferase activity in human tumors, cells, or xenografts. For example, the authors could express their LH2 mutant constructs in human lung cancer cells for orthotopic injection to demonstrate that human tumor cell metastasis is also sensitive to this novel LH2 function.

2. Secondly, and perhaps more seriously the authors have chosen an approach to lung cancer metastasis modeling that I have not seen before and which may not be appropriate. Simply counting the nodules found on the surface of a tissue type seems potentially misleading and incomplete. Generally speaking, metastatic burden is determined by immunohistochemistry of serial tissue sections. Nodules are counted and their size measured through the organ being evaluated. Alternatively, micro CT or other imaging approaches can more thoroughly address the question of metastatic burden coupled with rigorous quantification methods.

Can the authors justify their use of this approach? Do others consider this surface nodule counting method valid? Given that the differences in metastatic burden in figure 1 are somewhat modest, I am concerned that by ignoring the bulk of the tissue the authors have either blunted an exciting phenotype or have misled themselves.

Reviewer #2 (Remarks to the Author):

This manuscript presents data that LH2 has collagen telopeptide glycosyl transferase activity, and contributes to fibrosis and metastasis independent of its LH activity. Findings are novel and will be of interest to the field of extracellular matrix and cancer biology. However, some conclusions are overstated and not fully supported by data, and a few clarifications are required.

1. In introductory comments, clarification is required. Lysyl hydroxylase activity does not directly lead to cross-link formation. Lysyl oxidase activity is required for aldehyde formation, which is required. Whether or not the lysine residue is hydroxylated can influence the efficiency and final structure of cross-links in the microenvironment. Please revise accordingly.

2. Site directed mutagenesis studies could indicate the importance of electrostatic interactions between LH2 and collagen as suggested, but alternative explanations independent of electrostatic interactions between collagen and LH enzymes are also possible, such as indirect effects on conformation of LHs. The conclusions as stated do not allow for alternative possible interpretations.

3. Was complete loss of LH enzyme activity in tLH mutant enzyme with lower binding of iron directly confirmed in LH enzyme activity measurements (Figure 2)? If not, the data do not support claims made that metastatic contributions of LH2 are due exclusively to glycosyltransferase activity. In Figure 2j, the identity of EP-LH2 is not defined, or very easy to find.

4. The expected efficiency of cross-link formation from a glycosylated hydroxylysine residue would be expected to be diminished compared to a non-glycosylated hydroxylysine residue, due to the consequent lower reactivity of the epsilon amino group for both lysyl oxidase activity, and for cross-

link formation if unmodified by lysyl oxidases. Thus, the presence of glycosyl transferase activity from LH2 would be expected to reduce fibrosis and metastasis, rather than enhance it. The authors should directly comment on this question and their cross-link data from tumors in this context in the discussion. Collagen cross-linking in tumors would have to have been determined and compared as a function of LH2 splicing mutants in in vivo tumor experiments to answer this important question and apparent inconsistency with the premise of this manuscript.

Reviewer #3 (Remarks to the Author):

This paper describes the characterization of an isoform of Lysyl Hydroxylase 2 (LH2) and its impact on lung adenocarcinoma (LUAD) progression. Prior data had identified three LH isoforms (LH1, LH2, and LH3) that are each multidomain proteins that contain LH and glycosyltransferase (GT) domains with a bridging accessory (AC) domain of unknown function. Prior biochemical and structural data have indicated that LH3 contains not only LH activity, but also collagen Gal-OH-Lys glucosyltransferase activity (GGT activity). No GGT activity had been previously ascribed to LH1 and LH2. The present study identifies elevated expression of LH2 in lung adenocarcinoma and correlates that elevation with promoting metastasis. Their studies demonstrate electrostatic interactions between LH2 and collagen that provides unique telopeptidyl lysyl hydroxylase (tLH) activity that contributes to LUAD, but does not fully account for the metastatic promoting activity. Characterization of the GT domain of LH2 revealed that the expression of a splice isoform, termed LH2b, correlates with LUAD phenotypes and that LH2b has elevated GGT activity compared with the LH2a isoform. Deletion of the LH2b splice isoform inhibits LUAD and the authors propose that the LH2b GGT activity plays a significant role in lung tumor growth and metastasis.

Overall this is a well-written manuscript that comprehensively characterizes the role of LH2 in LUAD. However, several concerns must be addressed in revision of the manuscript including:

- 1) Abstract and the first paragraph of the discussion indicate that "protein crystallographic studies contributed to identification of collagen GGT activity for LH2b". In reality, the only X-tal structure presented in the manuscript is for LH3A1 and this added essentially nothing to the prior knowledge in the literature. The structure of LH3 was published by the Forneris lab in 2018. If the authors desire to include their LH3 structure in the present manuscript they at least need to make a comprehensive comparison with the prior PDB structure. However, the present manuscript does not even make much use of their LH3 structure except for homology modeling in Fig 3o (which needs to be described in the Methods). Surprisingly, the initial modeling of the LH domain in Fig 2h was instead based on the mimivirus LH structure rather than their own or the previously published LH3 structure. Subsequent modeling of the exon 13A loop used the Forneris published LH3 structure. Why is the new structure of LH3 presented in this paper? Where does it meaningfully add to the manuscript?
- 2) Abstract, Results and Discussion state that the structure "identifies an LH2 isoform (LH2b) that represents a novel glycosyltransferase (GLT) fold." In the first place, the fold is not novel if it is a homology model based on the structure of a previously published structure. Second, it is certainly not novel, because it is a classical metal-dependent GT-A fold as previously described by the Forneris lab. Third, the only 'novelty' of the structure is that the adjacent accessory domain has a loop insertion in the LH2b splice isoform that 'reaches back' and presumably interacts with the GT acceptor binding subsite. At this point all of the structural interpretation on LH2b is conjecture based on homology modeling and is certainly not a valid basis for ascribing a novel fold. There are numerous other GTs that have accessory domains that influence substrate interactions (e.g. GALNTs for binding O-glycosylated peptide substrates, POMGNT1 for binding glycopeptides for subsequent modification (both are GT-A folds), FUT8 with an adjacent C-terminal SH3 domain that contributes to substrate interactions (FUT8 is a GT-B fold enzyme). The contribution of an accessory domain that influences GT domain substrate interactions does not constitute a novel fold.
- 3) The legends to almost all of the figures are so terse and cryptic that they lack sufficient detail to

interpret what is being shown. All of the figure legends should be expanded to explain the respective panels and what is being measured without an undue burden of acronyms that make it challenging for a novice reader to understand.

4) The Results state: "By microscale thermophoresis, the UDP-glucose-binding activity of LH2b was higher than that of LH2a (Fig. 3p), and the isoforms demonstrated distinct binding modes (Extended Fig. 8)." Please provide controls that this fluorescein-tagged donor analog binds in the active site. Is it competitive with donor binding in enzyme assays?

Overall, the manuscript provides a compelling case for the roles of the LH, GT, and AC domains of LH2b in contributing to metastatic and growth promoting aspects of LUAD. The authors need to address some editing issues in the manuscript, but no additional experimentation is required. As a result, there is a high degree of enthusiasm regarding the manuscript if the revised submission can address these concerns.

Reviewer #1

1. All cell line and in vivo work employ a genetically engineered model of lung cancer in the mouse.

It would be helpful to see validation of the physiological relevance of LH2 glucosyltransferase activity in human tumors, cells, or xenografts. For example, the authors could express their LH2 mutant constructs in human lung cancer cells for orthotopic injection to demonstrate that human tumor cell metastasis is also sensitive to this novel LH2 function.

To address this question, we used CRISPR/Cas-9 gene editing to delete PLOD2 exon 13A in H358 lung cancer cells, which have detectable lysyl hydroxylase 2a (LH2a) and LH2b expression. Parental and two clones of exon 13A-deficient H358 cells were injected orthotopically into nu/nu mice. Mice were sacrificed 7 weeks after tumor cell injection to allow sufficient time for metastases to occur. Compared to parental H358 cells, exon 13A-deficient cells generated similar numbers of primary tumors and fewer metastases to the mediastinal lymph nodes and contralateral lung, but differences in metastasis numbers between parental and exon 13a-deficient tumors reached significance for only one of the mutant clones (see figure 1 below). Importantly, metastases generated by parental H358 cells were far fewer in number than they were in the syngeneic, immunocompetent tumor model, which reduced our ability to detect an effect of exon 13a deficiency in the context of H358 cells. Given that H358 tumors were generated in an immunodeficient host, these findings are in line with evidence presented here that high LH2 expression inhibits anti-tumor immunity and raise the possibility that an intact immune system is required for LH2-driven tumor progression. However, because of the caveats discussed above, we consider this result to be inconclusive and prefer not to include these data in the paper.

Figure 1: Numbers of visible primary lung tumors (left plot) and metastases to mediastinal nodes (middle plot) or contralateral lung (right plot) per mouse (dots). Mice were injected intra-thoracically with parental (WT) or LH2 exon 13A-deficient H358 cells and necropsied 7 weeks later.

2. Secondly, and perhaps more seriously the authors have chosen an approach to lung cancer metastasis modeling that I have not seen before and which may not be appropriate. Simply counting the nodules found on the surface of a tissue type seems potentially misleading and incomplete. Generally speaking, metastatic burden is determined by immunohistochemistry of serial tissue sections. Nodules are counted and their size measured through the organ being evaluated. Alternatively, micro CT or other imaging approaches can more thoroughly address the question of metastatic burden coupled with rigorous quantification methods.

Can the authors justify their use of this approach? Do others consider this surface nodule counting method valid? Given that the differences in metastatic burden in figure 1 are somewhat modest, I am concerned that by ignoring the bulk of the tissue the authors have either blunted an exciting phenotype or have misled themselves.

In the orthotopic lung tumor model reported here, tumor cells metastasize via the lymphatics to mediastinal lymph nodes and via the vasculature to the contralateral lung and other organs. Tumor cells that travel to the contralateral lung lodge in alveolar capillary beds in the lung periphery, generating metastatic deposits that are visible on the visceral pleural surface (PMID: 15999057), which recapitulates pleural metastases observed in lung adenocarcinoma patients (PMID: 23204236). Therefore, quantification of visible lung nodules on the pleural surface is a reasonable approximation of contralateral lung metastases in this model. Moreover, we have reported this quantitative approach widely (PMIDs: 19759262; 21403400; 22850877; 24762440; 25664850; 27869652; 29410444; 31969487).

Reviewer #2

1. In introductory comments, clarification is required. Lysyl hydroxylase activity does not directly lead to cross-link formation. Lysyl oxidase activity is required for aldehyde formation, which is required. Whether or not the lysine residue is hydroxylated can influence the efficiency and final structure of cross-links in the microenvironment. Please revise accordingly.

We revised the introduction to include the role of lysyl oxidases in collagen cross-link formation and the impact of lysine hydroxylation on collagen cross-link efficiency and structure.

2. Site directed mutagenesis studies could indicate the importance of electrostatic interactions between LH2 and collagen as suggested, but alternative explanations independent of electrostatic interactions between collagen and LH enzymes are also possible, such as indirect effects on conformation of LHs. The conclusions as stated do not allow for alternative possible interpretations.

We appreciate this point and cannot exclude the alternative possibility that site-directed mutagenesis led to unintended protein conformation changes that modulate collagen telopeptidyl lysyl hydroxylase (tLH) activity. Therefore, we have added the following statement to the Discussion section. "Finally, by performing domain swapping and rescue experiments on LH2-deficient osteoblasts, we showed that LH2's LH domain determines its tLH activity. Site-directed mutagenesis and enzymatic activity assays further identified the residues that are critical for LH2's tLH activity. These results indirectly suggest that a unique arginine cluster near LH2 active site may directly engage acidic residues in collagen telopeptides to facilitate substrate recognition. However, protein crystallographic studies will be required to substantiate this possibility."

3. Was complete loss of LH enzyme activity in tLH mutant enzyme with lower binding of iron directly confirmed in LH enzyme activity measurements (Figure 2)? If not, the data do not support claims made that metastatic contributions of LH2 are due exclusively to glycosyltransferase activity. In Figure 2j, the identity of EP-LH2 is not defined, or very easy to find.

We previously used purified recombinant proteins to show that LH2D689A mutation leads to loss of Fe²⁺-binding and tLH activities (PMIDs: 28216326; 29410444). We have modified the figure legend and text to define LH2-EP in Figure 2j.

4. The expected efficiency of cross-link formation from a glycosylated hydroxylysine residue would be expected to be diminished compared to a non-glycosylated hydroxylysine residue, due to the consequent lower reactivity of the epsilon amino group for both lysyl oxidase activity, and for cross-link formation if unmodified by lysyl oxidases. Thus, the presence of glycosyl transferase activity from LH2 would be expected to reduce fibrosis and metastasis, rather than enhance it. The authors should directly comment on this question and their cross-link data from tumors in this context in the discussion. Collagen cross-linking in tumors would have to have been determined and compared as a function of LH2 splicing mutants in in vivo tumor experiments to answer this important question and apparent inconsistency with the premise of this manuscript.

With respect to the reviewer's contention that lysyl oxidase enzymatic efficiency is expected to be diminished in the context of a glycosylated hydroxylysine (Hyl) residue on collagen, Hyl residues in collagen telopeptides are not glycosylated, so the relative "efficiency" of lysyl oxidases on glycosylated Hyl does not apply here. Furthermore, there are two helical sites in type I collagen where (glycosylated) Hyl can be involved in cross-linking, i.e. $\alpha 1(\alpha 2)$ -87 (near N-terminus) and $\alpha 1$ -930/ $\alpha 2$ -933 (near C-terminus). The Hyl residue at the latter site is not glycosylated (PMIDs: 24958722; 31173582). The Hyl residue at the former site ($\alpha 1(\alpha 2)$ -87) is the major glycosylation site in type I collagen; in many tissues, essentially all of the immature cross-links formed at this site are glycosylated (PMIDs: 22573318; 24958722; 31173582). We also reported that, in the periodontal ligament that has an exceptionally high rate of collagen turnover, the C-telo-Hyl/Lys residues are stoichiometrically converted to aldehyde and cross-linked to the $\alpha 1(\alpha 2)$ -87-Hyl on the neighboring molecule. Importantly, these cross-links are fully glycosylated, indicating that the glycosylated Hyl at $\alpha 1(\alpha 2)$ -87 is efficiently incorporated into cross-links (PMID: 3768322). Other reports also showed that the major cross-links derived from Hyl at this site are glycosylated (PMIDs: 8900119; 18442706). These data strongly indicate that the glycosylated Hyl residues actively and efficiently participate in cross-link formation. What we have found in the past was that the "di"-glycosylation of immature cross-links delay/prevent their "maturation" into tri-valent, mature cross-links (PMIDs: 22573318; 24958722) but not "cross-link formation".

Reviewer #3

1. Abstract and the first paragraph of the discussion indicate that "protein crystallographic studies contributed to identification of collagen GGT activity for LH2b". In reality, the only X-tal structure presented in the manuscript is for LH3A1 and this added essentially nothing to the prior knowledge in the literature. The structure of LH3 was published by the Forneris lab in 2018. If the authors desire to include their LH3 structure in the present manuscript they at least need to make a comprehensive comparison with the prior PDB structure. However, the present manuscript does not even make much use of their LH3 structure except for homology modeling in Fig 3o (which needs to be described in the Methods). Surprisingly, the initial modeling of the LH domain in Fig 2h was instead based on the mimivirus LH structure rather than their own or the previously published LH3 structure. Subsequent modeling of the exon 13A loop used the Forneris published LH3 structure. Why is the new

structure of LH3 presented in this paper? Where does it meaningfully add to the manuscript?

As suggested, we have removed the LH3A1 structure from the manuscript. For LH2 telopeptidyl LH homology modeling, we used the viral LH domain because it is a tLH (PMID: 29410444), whereas LH3 is not a tLH.

2. Abstract, Results and Discussion state that the structure “identifies an LH2 isoform (LH2b) that represents a novel glycosyltransferase (GLT) fold.” In the first place, the fold is not novel if it is a homology model based on the structure of a previously published structure. Second, it is certainly not novel, because it is a classical metal-dependent GT-A fold as previously described by the Forneris lab. Third, the only ‘novelty’ of the structure is that the adjacent accessory domain has a loop insertion in the LH2b splice isoform that ‘reaches back’ and presumably interacts with the GT acceptor binding subsite. At this point all of the structural interpretation on LH2b is conjecture based on homology modeling and is certainly not a valid basis for ascribing a novel fold. There are numerous other GTs that have accessory domains that influence substrate interactions (e.g. GALNTs for binding O-glycosylated peptide substrates, POMGNT1 for binding glycopeptides for subsequent modification (both are GT-A folds), FUT8 with an adjacent C-terminal SH3 domain that contributes to substrate interactions (FUT8 is a GT-B fold enzyme). The contribution of an accessory domain that influences GT domain substrate interactions does not constitute a novel fold.

We appreciate this point and have removed this conclusion from the Abstract, Results, and Discussion.

3. The legends to almost all of the figures are so terse and cryptic that they lack sufficient detail to interpret what is being shown. All of the figure legends should be expanded to explain the respective panels and what is being measured without an undue burden of acronyms that make it challenging for a novice reader to understand.

We have expanded the figure legends to improve their readability.

4. The Results state: “By microscale thermophoresis, the UDP-glucose-binding activity of LH2b was higher than that of LH2a (Fig. 3p), and the isoforms demonstrated distinct binding modes (Extended Fig. 8).” Please provide controls that this fluorescein-tagged donor analog binds in the active site. Is it competitive with donor binding in enzyme assays?

We performed the competition assay and found that unlabeled UDP-Glucose competes with fluorescein-tagged UDP-Glucose for binding to LH2 with an IC₅₀ of 30 μM.

REVIEWERS' COMMENTS:

Reviewer #1 (Remarks to the Author):

The authors rebuttal points are valuable and well taken. Specifically, their new experiment (figure 1, rebuttal) supports their hypothesis that LH2-mediated metastasis requires T regs and immunosuppressive macrophages to prevent CD8+ T cell recruitment. However, I disagree that they should not include their new data in the manuscript. Whereas the experiment would have been stronger if they had included control murine lung adenocarcinoma cells in their experiment to demonstrate that the murine cells exhibit the same phenotype, namely that they dont metastasize very well in an immunocompromised mouse lacking T-regs, the data are still valuable. i believe they should be included in the supplement as they may benefit readers and subsequent research.

Reviewer #3 (Remarks to the Author):

This is a revised version of a paper that describes the characterization of an isoform of Lysyl Hydroxylase 2 (LH2) and its impact on lung adenocarcinoma (LUAD) progression. The prior submission was considered to be well-written and it comprehensively characterized the role of LH2 in LUAD. However, there were several concerns regarding the inclusion of a crystal structure for LH3A1, conclusions regarding the "novelty" of the structural fold for the "GLT" domain, the terse nature of the figure legends, and the inclusion of appropriate controls for the thermophoresis experiments. The revised manuscript addressed most of these concerns, but one passage in the Discussion remains an ongoing concern regarding the potential conclusions that LH2b constitutes a novel fold.

The authors state: "LH2b seems to defy these classifications; its metal dependence and active site are GT-A-like, but its second Rossmann domain, which contains a loop that regulates co-substrate-binding and GGT activities, is GT-B-like (Extended Fig. 12d). Therefore, additional studies are warranted to determine whether LH2b represents a novel GLT fold."

Unfortunately, this statement and the corresponding Fig. 12d would lead a reader to conclude that there may be a question regarding the novelty of the GT fold type. There is absolutely no question in regard to the GT fold for LH2b based on the modeling shown in Fig. 3n. The homology model for the LH2b structure was based on the structure of human LH3. The GT domain of this protein is a classical GT-A fold. Not "maybe novel", it is absolutely, positively a classical GT-A fold. Yes, there is a second adjacent domain (accessory domain) comprised of a Rossmann-like fold, and a third LH domain. However, the features of the LH3 GT domain (and presumably the LH2b domain by homology) conform to the hallmark features of a GT-A fold (see PMID: 32234211). The presence of an additional adjacent domain that contributes to acceptor interaction does not qualify LH2b for reclassification of the fold as "novel". It is not, nor is the LH3 GT domain. As explained in the prior critique, there are many other GT-A and GT-B fold proteins that have accessory domains that contribute to substrate interactions. They are not reclassified as "novel" as a consequence of these extra domains even if they were Rossmann-like in their fold. It will perpetuate a misconception in the literature if Fig 12d and the statement in the Discussion are retained in this paper. They should be removed along with any mention of novelty in the protein fold.

Overall, the manuscript the manuscript is greatly improved. Only one additional round of editing is required to remove the reference to reference to 'novelty' of the GT fold and Fig 12d and the paper will be acceptable for publication.

Reviewer #1

Reviewer #1 (Remarks to the Author):

The authors rebuttal points are valuable and well taken. Specifically, their new experiment (figure 1, rebuttal) supports their hypothesis that LH2-mediated metastasis requires T regs and immunosuppressive macrophages to prevent CD8+ T cell recruitment. However, I disagree that they should not include their new data in the manuscript. Whereas the experiment would have been stronger if they had included control murine lung adenocarcinoma cells in their experiment to demonstrate that the murine cells exhibit the same phenotype, namely that they dont metastasize very well in an immunocompromised mouse lacking T-regs, the data are still valuable. i believe they should be included in the supplement as they may benefit readers and subsequent research.

Thanks for the suggestion. We included the new result as Figure 4I and provided a discussion about it.

2. Reviewer #3 (Remarks to the Author):

This is a revised version of a paper that describes the characterization of an isoform of Lysyl Hydroxylase 2 (LH2) and its impact on lung adenocarcinoma (LUAD) progression. The prior submission was considered to be well-written and it comprehensively characterized the role of LH2 in LUAD. However, there were several concerns regarding the inclusion of a crystal structure for LH3A1, conclusions regarding the “novelty” of the structural fold for the “GLT” domain, the terse nature of the figure legends, and the inclusion of appropriate controls for the thermophoresis experiments. The revised manuscript addressed most of these concerns, but one passage in the Discussion remains an ongoing concern regarding the potential conclusions that LH2b constitutes a novel fold.

The authors state: “LH2b seems to defy these classifications; its metal dependence and active site are GT-A-like, but its second Rossmann domain, which contains a loop that regulates co-substrate-binding and GGT activities, is GT-B-like (Extended Fig. 12d). Therefore, additional studies are warranted to determine whether LH2b represents a novel GLT fold.”

Unfortunately, this statement and the corresponding Fig. 12d would lead a reader to conclude that there may be a question regarding the novelty of the GT fold type. There is absolutely no question in regard to the GT fold for LH2b based on the modeling shown in Fig. 3n. The homology model for the LH2b structure was based on the structure of human LH3. The GT domain of this protein is a classical GT-A fold. Not “maybe novel”, it is absolutely, positively a classical GT-A fold. Yes, there is a second adjacent domain (accessory domain) comprised of a Rossmann-like fold, and a third LH domain. However, the features of the LH3 GT domain (and presumably the LH2b domain by homology) conform to the hallmark features of a GT-A fold (see PMID: 32234211). The presence of an additional adjacent domain that contributes to acceptor interaction does not qualify

LH2b for reclassification of the fold as “novel”. It is not, nor is the LH3 GT domain. As explained in the prior critique, there

are many other GT-A and GT-B fold proteins that have accessory domains that contribute to substrate interactions. They are not reclassified as “novel” as a consequence of these extra domains even if they were Rossmann-like in their fold. It will perpetuate a misconception in the literature if Fig 12d and the statement in the Discussion are retained in this paper. They should be removed along with any mention of novelty in the protein fold.

Overall, the manuscript the manuscript is greatly improved. Only one additional round of editing is required to remove the reference to reference to ‘novelty’ of the GT fold and Fig 12d and the paper will be acceptable for publication.

We removed the discussion on the novel fold as suggested by the reviewer.